# Achievements and Challenges in the Prevention of Mother-to-Child Transmission of HIV—A Retrospective Cohort Study from a Rural Hospital in Northern Tanzania

**DOI:** 10.3390/ijerph18052751

**Published:** 2021-03-09

**Authors:** Sunniva Marie Nydal, Yuda Munyaw, Johan N. Bruun, Arne Broch Brantsæter

**Affiliations:** 1Institute of Clinical Medicine, Faculty of Medicine, University of Oslo, P.O. Box 1171, Blindern, 0318 Oslo, Norway; sunnivanydal@gmail.com (S.M.N.); j.n.bruun@medisin.uio.no (J.N.B.); 2Department of Obstetrics and Gynecology, Haydom Lutheran Hospital, P.O. Box 9000, Haydom, Mbulu Manyara, Tanzania; munyaw@rocketmail.com; 3Department of Infectious Diseases, Ullevål Hospital, Oslo University Hospital, P.O. Box 4956, Nydalen, 0424 Oslo, Norway; 4Department of Acute Medicine, Ullevål Hospital, Oslo University Hospital, P.O. Box 4956, Nydalen, 0424 Oslo, Norway

**Keywords:** prevention of mother-to-child transmission of HIV (PMTCT), HIV-exposed infants, pregnancy, antiretroviral therapy, Tanzania

## Abstract

Despite the goal of eliminating new human immunodeficiency virus (HIV) infections in children, mother-to-child transmission is still common in resource-poor countries. The aims of this study were to assess the occurrence of mother-to-child transmission of HIV (MTCT) by age 18 months, risk factors for transmission, and the implementation of the national prevention of MTCT (PMTCT) program in a rural hospital in Tanzania. Data were collated from various medical registers and records. We included 172 children and 167 HIV-infected mothers. Among 88 children (51%) with adequate information, 9 (10.2%) were infected. Increased risk of MTCT was associated with late testing of the child (>2 months) [OR = 9.5 (95% CI: 1.8–49.4)], absence of antiretroviral therapy during pregnancy [OR = 9.7 (95% CI: 2.1–46.1)], and maternal CD4 cell count <200 cells/mm^3^ [OR = 15.3 (95% CI: 2.1–111)]. We were unable to determine the occurrence of MTCT transmission in 84 children (49%). The results from this study highlight that there is an urgent need for enhanced efforts to improve follow-up of HIV-exposed children, to improve documentation in registries and records, and to facilitate ease of linkage between these.

## 1. Introduction

In 2015, the Sustainable Development Goals set out the ambitious targets of ending the acquired immunodeficiency syndrome (AIDS) epidemic and achieving universal health coverage by 2030 [1]. This was followed by the goal of eliminating new human immunodeficiency virus (HIV) infections in children by 2018 [2]. Yet, in 2019, 150,000 children under 15 years became infected with HIV. The majority of these infections occurred in Sub-Saharan Africa, and in Tanzania, there were an estimated 6300 cases of new HIV infections in this age group [3].

The primary cause of HIV infection in children is vertical transmission, accounting for over 90% of infections in children under 15 years [4]. The risk of mother-to-child transmission of HIV (MTCT) from an untreated mother ranges from 15–45%. Breastfeeding increases the risk of MTCT [5], but in resource-poor countries this risk is generally counterbalanced by prevention of other severe infections.

In resource-rich settings, prevention of MTCT (PMTCT) interventions including antiretroviral therapy (ART) in pregnancy, use of caesarean section in mothers with HIV viraemia, and avoidance of breastfeeding have reduced the risk of MTCT to less than 2% [6]. Similar results have also been demonstrated in some studies from Africa [7,8,9]. However, UNAIDS reported a vertical transmission rate of 11% in Tanzania in 2019 [3].

The PMTCT cascade is a series of key stepwise activities, starting with diagnosis and treatment of all pregnant women, continuing with newborn antiretroviral prophylaxis, and ending with the determination of HIV status of HIV-exposed infants (HEIs) at 18 months of age [10]. 

In Tanzania, PMTCT activities include regular HIV-testing during pregnancy and at the time of delivery, and provision of lifelong ART for all HIV-infected women. Antiretroviral (ARV) prophylaxis is recommended for HEIs during the first 6 weeks of life. In addition, cotrimoxazole preventive therapy (CPT) is recommended for newborns from age 4–6 weeks and until HIV infection has been ruled out after complete cessation of breastfeeding. Exclusive breastfeeding is recommended for the first 6 months, followed by mixed feeding until 12 months, before cessation of breastfeeding by month 13. Other measures include family planning and referral of HIV-infected infants to Care and Treatment Clinics (CTCs) [4]. 

Although PMTCT activities are in place in most resource-constrained countries, there is a need for studies that examine in depth the level of implementation and quality of documentation at the health facility level, and how this may affect the reported rate of MTCT. 

The aims of our study were to assess the occurrence of MTCT by age 18 months, the influence of known predictors on transmission, and the implementation of PMTCT measures as described in the Tanzanian national guidelines. 

## 2. Materials and Methods

### 2.1. Participants and Setting

Haydom Lutheran Hospital is a referral hospital in rural northern Tanzania which has provided ART free of charge since October 2003 [11].

We included mothers and HEIs enrolled in the postnatal PMTCT program at the Reproductive and Child Health Service (RCHS) from January 2014 to December 2016. The children were followed up to age 18 months, or to December 2018 in the case of HIV-infected children. Nine mother-infant pairs were excluded due to lack of unique identity (ID)-numbers (see “Data collection”). In total, 172 children born to 167 mothers were included (Figure 1) in this retrospective cohort study.

### 2.2. Laboratory Testing for HIV Infection in Mothers and Children

All mothers not already known to be HIV-infected were screened for HIV infection as part of antenatal care using a finger prick rapid antibody test (rapid test) (BIOLINE HIV-1/2 3.0 test kit, Standard Diagnostics Inc., Suwon-si, Korea). For confirmation of a positive screening test, a different rapid test was used (Uni-Gold HIV-1/2, Trinity Biotech, Bray, Ireland) [4]. In children, MTCT was diagnosed by HIV DNA polymerase chain reaction (PCR) and rapid tests. According to Tanzanian guidelines, all HEIs should have a HIV DNA PCR test at 4–6 weeks of age, a repeat test 6 weeks after cessation of breastfeeding, or at any time of symptoms [4]. Passively transferred antibodies from mother-to-child diminish rapidly between 9–18 months of age. Therefore, the type of recommended repeat test depends on the age of the child: an HIV DNA PCR test is used under 9 months, and a rapid test after the 9 months. A rapid test is recommended at 18 months to confirm HIV negative status. HIV DNA PCR analysis is not performed at our hospital but dried blood spot (DBS) samples from a finger, heel or big toe prick are transported to the Kilimanjaro Christian Medical Centre Clinical Laboratory for analysis. The average turnaround time is around two weeks. 

### 2.3. Categorization of Children Based on Laboratory Test Results

The children were followed until age 18 months and were finally categorized into one of three groups: “HIV-infected”, “HIV-uninfected”, and “HIV status indeterminable”. Reasons for being included in the latter group included no or inadequate testing (in relation to birth and breastfeeding), transfer out to other health care providers, loss to follow-up (for unknown reason), and death without appropriate HIV testing. Any positive HIV DNA PCR test or rapid test was accepted as evidence of HIV infection in the child, even in the case of missing evidence of a confirmatory test result. We accepted a negative rapid test after cessation of breastfeeding as indication of absence of mother-to-child transmission (MTCT), even with poor documentation of timing in relation to cessation of breastfeeding, and also when there was no record of rapid test having been performed at age 18 months.

### 2.4. Sources and Linkage of Data

We used routinely collected data from hospital records of PMTCT activities. To ensure deidentification of participants, we assigned a study-specific identification code to each mother-child pair and recorded no person-identifiable information in the data file. Laboratory and clinical data came from three separate health care providers at Haydom Lutheran Hospital: Reproductive and Child Health Service (RCHS):
The “The HIV-exposed child follow-up register” which contains data on the mother’s use of ARVs during pregnancy, and child data up to age 18 months, including date of birth, birth weight, gender, receipt of nevirapine prophylaxis, CPT, feeding regimen, date and results of HIV DNA PCR and rapid tests, and CTC number of the child.A separate logbook with the names of mothers and their children’s HIV early infant diagnosis numbers.Care and Treatment Clinic (CTC):The medical files of patients.The electronic database which contains laboratory and clinical data on HIV-infected individuals. When available, data in “the HIV-exposed Infant Cards” embedded in the mothers’ CTC files were also used. These cards contain brief summaries of the child’s PMTCT history, i.e., dates of HIV testing and mode of infant feeding.
Labor ward: the labor and delivery registers which provides details on mode of delivery.

In the HIV-exposed child follow-up register (Figure 2), data are deidentified both for mothers and their children but contain columns for recording CTC file number and CTC identity (ID) numbers. When recorded, this enabled linkage to records at the CTC. In case of missing CTC file number, but available CTC ID number, the latter was used to find the patient’s file number by means of the CTC database. If both ID numbers and file numbers were missing, an attempt was made to identify the mother via the child’s HIV early infant diagnosis number found in the HIV-exposed child follow-up register. A separate RCHS logbook links these numbers to the mother’s name. When this was successful, the ID numbers and CTC file numbers were retrieved from the data base by a name search. The HIV early infant diagnosis numbers and names of mothers were also used to link the patients to the labor ward logbooks. 

### 2.5. Statistical Methods

Data were entered into an Excel spread sheet and exported to SPSS for statistical analysis. The significance level was set at 5%, and all analyses were performed using IBM^®^ SPSS^®^ Statistics version 25 for Windows (IBM Corp., Armonk, NY, USA).

Baseline characteristics of all children and mothers enrolled in the study are described as proportions for categorical data and as median and interquartile range (IQR) for continuous variables. We examined differences in proportions between groups using Pearson’s chi-square test, and differences in continuous variables between groups using non-parametric Mann–Whitney U-test. 

For the purpose of analysis of the effect of late HIV DNA PCR testing on the risk of MTCT, the cut-off value was set at child age two months instead of six weeks as is recommended by the national guidelines. Six weeks was an unfavorable cut-off value as only one child in the HIV-infected group had undergone HIV DNA PCR testing before six weeks of age.

Maternal and child characteristics relevant for MTCT were selected based on existing knowledge. We used binary logistic regression to assess the associations between child and maternal characteristics and the risk of MTCT. We present associations (risk of MTCT) as odds ratios (OR) with 95% Confidence Intervals (95% CI). Multivariable adjustment was not performed due to the low number of infected children.

## 3. Results

### 3.1. Determination of MTCT

An early HIV DNA PCR test was performed in 95.9% of the children (Table 1), but only 27.3% had this test at the recommended age of 4–6 weeks. Only 55.8% had a documented second test. We were unable to ascertain if transmission of HIV had occurred by age 18 months in 84 (48.8%) children (indeterminable group) due to inadequate testing, missing information, deaths, transfer out, or loss to follow-up. However, 92.9% in this group had an early HIV DNA PCR test with negative result.

Nine children (5.2%) tested HIV PCR positive in the total study population. However, among children with a determinable outcome at age 18 months, 10.2% were positive, eight by the first test, and one by a test performed after cessation of breastfeeding (4 months after). None of nine HIV-infected children, had a documented second HIV DNA PCR test to confirm infection. The proportion of children presenting late for HIV DNA PCR testing was highest among infected children. 

### 3.2. Characteristics of the Mothers

The median age of mothers at the time of childbirth was 29 years (Table 2). Vaginal delivery was most common, but in 82 pregnancies (47.7%) the mode of delivery was not recorded due to delivery at home or in another health facility. The median CD4 cell count around time of delivery was lowest for mothers of infected children, and these mothers also had more severe disease according to the WHO stage before delivery.

### 3.3. Characteristics of The Children and Retention in Care

Median birth weight was 3000 g for all children with available data, 3200 g for HIV-uninfected vs. 2500 for infected (Table 3). The median child age at RCHS enrolment was higher for HIV-infected than for non-infected children (13.1 vs. 6.7 weeks). The main reason for indeterminable HIV status at age 18 months were loss to follow-up and transfer to other health care facilities. Across all groups six children were known to have died.

Only five of nine HIV-infected children were referred to the CTC, and three of these were later transferred to other treatment facilities. Three (33.3%) HIV-infected children died, all without treatment. Time from date of HIV DNA PCR test to death was known for two of the fatalities and was more than 2 months in both cases.

### 3.4. ART in Pregnancy and ARV Newborn Prophylaxis

The large majority (87.2%) of the mothers received ART during pregnancy (Table 4), mostly as triple therapy comprising tenofovir, lamivudine, and efavirenz. Additionally, most infants (85.5%) received nevirapine prophylaxis. A lower proportion of infected children (55.6%) had mothers receiving ART than children in the uninfected group (92.4%). Similarly, coverage of nevirapine prophylaxis was lower in the infected group (77.8%) than in the uninfected group (91.1%).

### 3.5. Cotrimoxazole Preventive Therapy (CPT)

Nearly all infants (98.8%) started CPT, but the majority (62.9%) started after the recommended age 4–6 weeks, and later in infected (13.1 weeks) than in uninfected children (6.6 weeks) (Table 5). In the uninfected group, most (79.7%) continued CPT until six weeks after weaning or until HIV infection was ruled out. Among HIV-infected children, none had documentation of continued CPT during follow-up as recommended.

### 3.6. Infant Feeding

Across all groups there was poor documentation of breastfeeding practice (Table 6). Among HIV-uninfected children with adequate documentation, two-thirds were exclusively breastfed during the first 6 months, half received mixed feeding during the subsequent 6 months, and three-fourths had stopped breastfeeding by 13 months, as per guideline.

### 3.7. Factors Associated with MTCT

Eighty-eight infants had adequately documented HIV status at 18 months. Infants born to mothers who did not receive ART during pregnancy had significantly increased risk of infection OR = 9.7 (95% CI: 2.1–46.1) (Table 7). Children who presented late (>2 months) for HIV DNA PCR testing had significantly higher risk of infection than infants tested earlier, OR = 9.5 (95% CI: 1.8–49.4). Maternal CD4 cell counts <200 cells/mm^3^ was significantly associated with increased risk of MTCT, OR = 15.3 (95% CI: 2.1–111).

Data completeness for breastfeeding and birth weight were not adequate for analysis of risk of transmission to the child.

### 3.8. Characteristics of Mothers and Children with Indeterminable HIV Status at 18 Months

Mothers of children in the uninfected and indeterminable groups had similar coverage of ART during pregnancy (92.4% and 85.7%, respectively, *p* = 0.329). This was significantly higher than in the infected group (55.6%), *p* = 0.001 and *p* = 0.011, respectively (Table 4). Maternal median CD4 cell counts were similar in the uninfected and indeterminable group (488 cells/μL and 422 cells/μL, respectively, *p* = 0.387 (Table 2), again higher than in the infected group (231 cells/μL), *p* = 0.032 and *p* = 0.056, respectively. Additionally, when dividing the groups into categories of maternal CD4 cell count below and above 200 cells/mm^3^, there was no difference between the uninfected and indeterminable groups, *p* = 0.260.

Median infant age at first HIV DNA PCR test was similar in the uninfected and indeterminable groups (48 days and 46.5 days, respectively, *p* = 0.647), and they presented earlier than infected children (92 days), *p* = 0.004 (Table 1). Additionally, when dividing the groups into categories of child age at first HIV DNA PCR test below and above 2 months, there was no difference between the uninfected and indeterminable groups, *p* = 0.383.

Finally, the proportion of asymptomatic women (WHO stage I infection) was similar in the uninfected and indeterminable group groups (55.7% and 53.7%, respectively), *p* = 0.827 (Table 2). These proportions were higher than in the infected group (28.6%), but not significantly different (*p* = 0.173 and *p* = 0.211 for uninfected and indeterminable versus infected, respectively).

## 4. Discussion

MTCT is a major public health challenge in many resource-poor countries. We found that 10.2% of HEIs with known outcome at 18 months were infected. Factors associated with MTCT included mothers not taking ART in pregnancy, maternal CD4 cell count <200 cells/mm^3^, and late HIV DNA PCR testing of infants (>2 months). However, for almost one half of the children enrolled in the postnatal PMTCT program we were unable to ascertain if the children were infected, largely due to loss to follow-up and transfer out to other health care providers. Several weaknesses in the implementation of the PMTCT program were identified in relation to prevention and diagnosis of HIV infection and follow-up of exposed children.

The rate of MTCT in our study is in close agreement with the Tanzanian national rate of 11% in 2019 [3], and also with two earlier studies that found transmission rates of 9.6 and 10.5% [12,13]. None of the infected children in our study had a second confirmatory HIV PCR test. We therefore cannot exclude the occurrence of false-positive tests. A study from South Africa reported a positive predictive value (PPV) of 94.4% for the first HIV DNA PCR test result after confirmatory re-testing [14]. We therefore propose that greater emphasis be put on performing a confirmatory test, and that this is documented in the HIV-exposed child follow-up register.

Although we could not determine the HIV status at age 18 months in almost one half of the children, two factors suggest that the rate of transmission in this group is likely to be low. First, the indeterminable group was similar to the uninfected group and differed significantly from the infected group in terms of several known predictors of MTCT, including the mothers’ use of ART, her CD4 cell counts, and WHO stage. Second, studies conducted before the era of ART demonstrated that roughly one third of children were infected in utero, one third during birth, and a further one third by breastfeeding [15]. However, for women on effective ART the risk of transmission is very low, with a target of less than 5% transmission among breastfeeding women [5]. In the indeterminable group, 78 children (92.9%) had an early PCR test that was negative, practically excluding in utero and peripartum transmission. These factors suggest that the proportion of infected children in the indeterminable group is likely to be considerably lower than 10.2%. On the other hand, children lost to follow-up may be at increased risk of HIV infection from mothers that have stopped taking ART [16]. Furthermore, testing may have been performed less than 6 weeks after cessation of breastfeeding in some cases, as we accepted a negative rapid test as indication of absence of MTCT, even in the case of poorly documented number of weeks after stopping breastfeeding. Therefore, undetected MTCT may have occurred later. To ensure correct timing of testing after cessation of breastfeeding, we propose inserting an extra column specifically for “Date of cessation of breastfeeding” or “Number of weeks since cessation of breastfeeding at the time of HIV-testing”. On balance, we find it likely that the rate of transmission in the entire study population is between 10.2 % and 5.2%, most likely in the lower end of this range.

We found that not taking ART during pregnancy was significantly associated with the likelihood of MTCT. There was also a smaller and non-significant increase in risk of MTCT among children who did not receive infant ARV prophylaxis. Our results may indicate that maternal ART is a more important protective factor than infant ARV prophylaxis. This is in agreement with previous studies from Kenya, Zambia and Nigeria [17,18,19].

Early diagnosis, treatment, and retention in care of the women are challenges in our rural setting in Tanzania. Prescription of antiretroviral drugs, both for treatment of the mother and prophylaxis in the child, will only be effective if the mother is compliant with treatment and provision of prophylaxis to her infant. Additionally, HIV resistance may threaten the success of PMTCT activities. Other studies have demonstrated that resistance to non-nucleoside reverse transcriptase inhibitors such as nevirapine is common in HIV -infected infants in Tanzania [20,21]. Therefore, monitoring of resistance to all classes of ARVs is required both on population and individual level in order to prevent MTCT.

Only 27.3% of the children had a first HIV PCR test as recommended at 4–6 weeks of age. This is even lower than the 57.1% reported in another recent study from North-East Tanzania [22]. Testing after 2 months of age was associated with significantly higher odds of infection, in agreement with previous studies from Tanzania and Ethiopia [12,23]. The association between late testing and vertical transmission may be caused by older infants tending to present for diagnostic testing because of symptoms, as opposed to routine early infant screening. The association may also be due to confounding factors like increased risk behavior among mothers of late enrollers at RCHS. We propose that greater effort be placed on encouraging and reminding mothers to bring in their children at the age of 4–6 weeks. Additionally, we propose that HIV DNA PCR testing be introduced on site to prevent delays caused by slow turnaround time.

The odds of MTCT were over 15 times higher among mothers with CD4 cell counts <200 compared to mothers with higher levels. Similarly, a study from South Africa found that higher maternal baseline CD4 cell count was associated with reduced odds of MTCT [8]. Low CD4 cell counts may be a result of late initiation of treatment, antiviral resistance, or poor adherence to therapy caused by stigma, logistic challenges, or lack of understanding. Low CD4 cell counts probably reflect no or ineffective ART—resulting in high viral load and subsequent high risk of MTCT.

The WHO recommends CPT after week 4–6 to prevent bacterial infections, malaria and *Pneumocystis jirovecii* pneumonia [24]. We found that the majority of HEIs started CPT at a later age, possibly putting the children at risk of severe infections. However, the recommendation to give CPT to HIV-exposed, HIV-uninfected infants is based on very-low-quality evidence [24]. A recent study from South Africa suggests that co-trimoxazole prophylaxis provides no benefit after age 6 weeks for HIV-uninfected breastfed infants in countries that are unaffected by malaria [25]. Malaria is not highly prevalent in children seen at our hospital, and it is possible that that failure to provide CPT in HIV-uninfected infants may not be of great concern.

Only five of nine HIV-infected children (55.6%) in our study were referred to the CTC, and a previous study from Tanzania found that a mere one third of children with an early positive PCR test had documented referral to therapy [20]. The clinical course of HIV infection is often aggressive in children not offered ART. By one year, an estimated one third of children will have died, and by two years approximately one half [26,27]. Increased likelihood of malnourishment and illness among loss to follow-up infants compared to children retained in care [28], further contribute to the risk of death. Food supplementation has been found to be protective against loss to follow-up among HIV- exposed infants by acting as an incentive for continued pediatric care and treatment [29]. Food supplementation may also help reduce malnourishment, which is associated not only with loss to follow-up, but also with increased mortality. The WHO recommends integrating HIV prevention, care and treatment services with reproductive and child health services to reduce overall maternal and child mortality [30]. Stronger linkage between services would help identifying mothers and children at risk of loss to follow-up, and thereby improve retention in care.

Of all children enrolled in our study, one third (32.0%) was lost to follow-up for unknown reasons, and one-eighth (12.7%) was transferred out to other health care providers. This largely explains why we were unable to adequately determine the HIV status at age 18 months in almost one half of the exposed children. Long distances, cost of travel and poor communications in our rural area may be good reasons for transfer of patients to health care providers nearer to the home. Loss to follow-up for unknown reasons is a greater cause for concern and has also been identified as a significant challenge in several other studies from Africa [13,17,19,28,31]. In addition to the above factors, stigma associated with HIV is a likely explanation for the high rate of loss to follow-up.

Linkage of data and continuity of care between different health service providers within and outside the hospital is challenging. In our hospital, HIV-infected mothers and their children receive HIV-related services from RCHS, Labor Ward, and CTC. These are located in separate buildings and are staffed by separate groups of health care workers who enter data into different medical registers and records. The large amount of paperwork and frequent updates in reporting systems constitute a heavy workload for the staff that may result in failure to correctly and comprehensively document all required aspects of PMTCT. Our observations are in accordance with other studies from South Africa and Tanzania which have found that a multiplicity of registers contributes to unnecessary complexity and inaccuracy of data [32,33]. Lack of coordination and poor data quality may impede effective follow-up, treatment, and retention in care.

There is therefore a need for ongoing training of health care personnel in correct and complete documentation. In addition, we propose that regular auditing of logbooks and patient files be implemented. To strengthen the linkage between medical records at RCHS and CTC, we suggest that the mother’s CTC ID numbers be accurately recorded in all PMTCT registers and records, and that HEI numbers, test results, and infant prophylaxis be recorded in the mother’s CTC file. Reducing the number of PMTCT registers and records using a minimal common set of key indicators, as well as closer proximity of service locations, may facilitate follow-up and reduce paperwork for health workers [33,34]. Unfortunately, required care and treatment for HIV is still associated with stigma, and this continues to be an impediment to closer integration of CTC activities in the ordinary services of the hospitals in Africa.

The principal strength of this study is the “real world” rural setting and having collected information from a wide range of clinical records and registries that are routinely available in a hospital. By this approach, we have been able to identify and describe practical challenges in the postnatal PMTCT program in Tanzania. In contrast, most PMTCT studies are performed in well-resourced centers or are based on programmatic data. However, our study also has clear limitations, mainly related to the large group of children with incomplete data and indeterminable HIV status at 18 months. However, in the context of public health, this is an important finding that merits attention by health officials in order to improve care for HEIs. Another limitation is the relatively small study population in a single institution. The external validity of our study is therefore uncertain, but we have no reason to believe that the challenges identified in our hospital are of less concern in other rural hospitals. Finally, we did not have information on HIV viral load or resistance in sufficient individuals to include these parameters in our study.

## 5. Conclusions

One in ten HIV-exposed children with sufficient data for assessment were infected from their mothers by age 18 months. Risk of infection was associated with low maternal CD 4 cell counts, mothers not taking ART during pregnancy, and late presentation of infants for HIV testing. Almost one-half of the children enrolled at RCHS could not be adequately assessed for HIV infection, largely due to loss to follow-up and transfer out to other health care providers.

Our study demonstrates a need for enhanced efforts to prevent and diagnose MTCT, reduce loss to follow-up of HIV-exposed children, to improve documentation in registries and records, and to facilitate ease of linkage between these.

## Figures and Tables

**Figure 1 ijerph-18-02751-f001:**
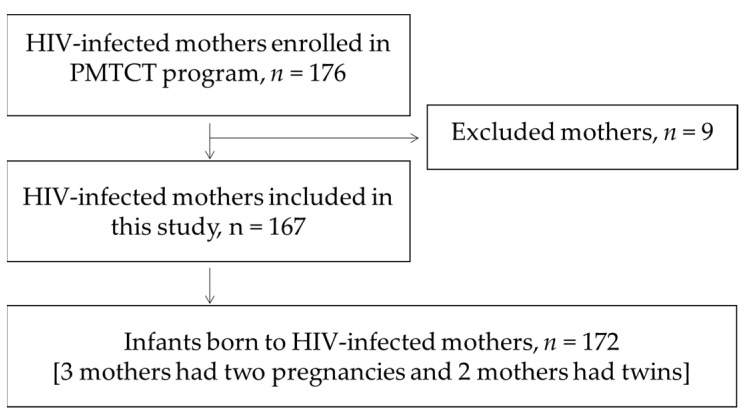
Flow chart of inclusion of mothers and children. PMTCT: Prevention of mother-to-child transmission of human immunodeficiency virus (HIV).

**Figure 2 ijerph-18-02751-f002:**
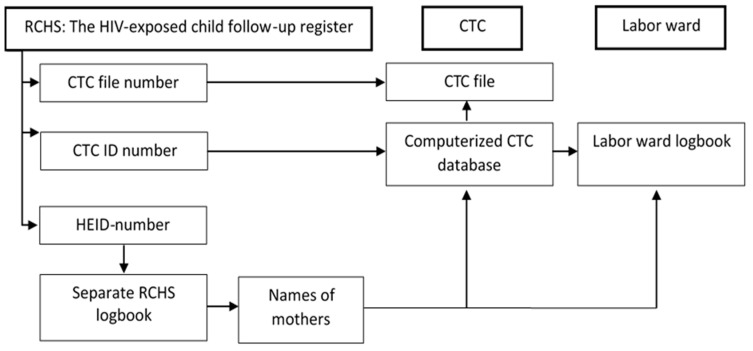
Linkage of data for mothers and their children between RCHS, CTC and labor ward. Abbreviations: RCHS: Reproductive and Child Health Service; CTC: Care and Treatment Clinic; HEID: HIV Early Infant Diagnosis.

**Table 1 ijerph-18-02751-t001:** Overview of diagnostic testing of children as basis for categorization of children by HIV status.

Diagnostic Test Information	HIV-Infected	HIV-Uninfected	HIV Status Indeterminable	Total
	*n* = 9	*n* = 79	*n* = 84	*n* = 172
**Number who had a first PCR test, *n* (%)**	9 (100)	78 (99.9)	78 (92.9)	165 (95.9)
PCR test done before 4 weeks	0	4 (5.1)	3 (3.6)	7 (4.1)
PCR test done at 4–6 weeks	1 (11.1)	21 (26.6)	25 (29.8)	47 (27.3)
PCR test done after 6 weeks	8 (88.9)	53 (67.1)	50 (59.5)	111 (64.5)
Missing data (did not have first PCR test)	0	1 (1.3)	6 (7.1)	7 (4.1)
**Timing of First PCR**				
PCR test done before 2 months	2 (22.2)	59 (74.7)	54 (64.3)	115 (66.9)
PCR test done after 2 months	7 (77.8)	19 (24.0)	24 (28.6)	50 (29.1)
Missing data	0	1 (1.3)	6 (7.1)	7 (4.0)
**Median age at first PCR (IQR), days**	92 (75–106)	48 (40.3–61)	46.5 (34.8–67.8)	48 (39–67)
**Number who had a second test, *n* (%)**				
Rapid test	NA ^1^	78 (98.7) ^3^	16 (19.0)	94 (57.6)
Median age in months (IQR)	-	16 (14–18)	15 (12.0–16.8)	16 (14–18)
PCR test	1 (11.1) ^2^	5 (6.3)	0	6 (3.5)
Median age in months (IQR)	16	10 (9–14)		12 (9.3–14.8)
Rapid test and PCR test	0	4 (5.1)	0	4 (2.3)
Missing data	8 (88.9)	0	68 (81.0)	76 (44.2)

^1^ Not applicable (NA) as children already diagnosed with HIV by PCR are not required to undergo repeat rapid testing after cessation of breastfeeding; ^2^ In addition to the 78 who had a rapid test performed at a correct time, one child had the PCR test only and four children had both tests; ^3^ One of the infected children had two PCR tests. The second test, done after cessation of breastfeeding, was reactive.

**Table 2 ijerph-18-02751-t002:** Maternal characteristics by child HIV status.

Maternal Characteristics	HIV-Infected	HIV-Uninfected	HIV Status Indeterminable	Total
	*n* = 9	*n* = 79	*n* = 84	*n* = 172
**Age in years at the time of delivery,**				
median (IQR) ^1^	31 (27–36)	28 (23–34)	30 (26.5–34.5)	29 (24–34)
**CD4 cell count (cells/mm^3^) nearest to time of delivery,**				
median (IQR) ^1^	231 (81–335)	488 (310–688)	422 (298–620)	434 (290–667)
**Days between CD4 cell count (cells/mm^3^) and delivery,**				
median (IQR) ^1^	206 (123–404)	109 (49–239)	138 (65–242)	129 (56–248)
**WHO stage before delivery, *n* (%) ^1^**				
Stage 1	2 (28.6)	34 (55.7)	29 (53.7)	65 (53.3)
Stage 2	1 (14.3)	5 (8.2)	9 (16.7)	15 (12.3)
Stage 3	1 (14.3)	9 (14.8)	4 (7.4)	14 (11.4)
Stage 4	3 (42.9)	13 (21.3)	12 (22.2)	28 (23.0)
**Mode of delivery, *n* (%) ^1^**				
Spontaneous vaginal delivery	2 (22.2)	42 (53.1)	33 (39.2)	77 (44.7)
C-section	0 (0)	9 (11.4)	4 (4.7)	13 (7.6)

^1^ Missing data: Age: 2 in the infected group, 18 in the uninfected, 29 in the indeterminable group; CD4 cell count and days between CD4 cell count and delivery: 3 in the infected group, 30 in the uninfected, 46 in the indeterminable group; WHO stage: 2 in the infected group, 18 in the uninfected, 30 in the indeterminable group; Mode of delivery: seven in the infected group, 28 in the uninfected, 47 in the indeterminable group.

**Table 3 ijerph-18-02751-t003:** Child characteristics and retention in care at the Reproductive and Child Health Service (RCHS) by HIV status.

Child Characteristics	HIV-Infected	HIV-Uninfected	HIV Status Indeterminable	Total
	*n* = 9	*n* = 79	*n* = 84	*n* = 172
**Sex, *n* (%) ^1^**				
Female	4 (44.4)	37 (48.1)	37 (45.1)	78 (46.4)
Male	5 (55.6)	40 (51.9)	45 (54.9)	90 (53.6)
**Birth weight in grams,**				
**median (IQR) ^1^**	2500 (2500–3000)	3200 (3000–3500)	3000 (2750–3220)	3000 (3000–3500)
**Age at enrolment in weeks,**				
**median (IQR)** **^1^**	13.1 (10.7–15.1)	6.7 (5.5–8.8)	6.6 (4.6–9.5)	6.7 (4.9–9.7)
**Retention in care at RCHS at age of 18 months, *n* (%)**				
Alive and in care at RCHS				
Discharged early with negative HIV	3 (33.3)	58 (73.4)	9 (10.7)	70 (40.7)
Test	0 (0)	19 (24.0)	0 (0)	19 (11.0)
Died	3 (33.3)	1 (1.3)	2 (2.4)	6 (3.5)
Transferred out	2 (22.2)	1 (1.3)	19 (22.6)	22 (12.7)
Lost to follow-up	1 (11.1)	0 (0)	54 (64.3)	55 (32.0)

^1^ Missing data: Sex: 2 in the uninfected group, 2 in the indeterminable group; Birth Weight: 4 in the infected group, 21 in the uninfected, 29 in the indeterminable group; Age at enrolment: 1 in the indeterminable group.

**Table 4 ijerph-18-02751-t004:** Antiretroviral therapy (ART) for mothers and nevirapine prophylaxis for children by child HIV infection status.

Use of Antiretrovirals	HIV-Infected	HIV-Uninfected	HIV Status Indeterminable	Total
	*n* = 9	*n* = 79	*n* = 84	*n* = 172
**Mothers**				
**Type of maternal intrapartum ART, *n* (%)**				
Any ART ^1^	5 (55.6)	73 (92.4)	72 (85.7)	150 (87.2)
TDF+3TC+EFV	5 (55.6)	72 (91.1)	68 (81.0)	145 (84.3)
Other ART (AZT or AZT+3TC+sdNVP)	0 (0)	1 (1.3)	4 (4.8)	5 (2.9)
None	4 (44.4)	6 (7.6)	10 (11.9)	20 (11.6)
Missing data	0	0	2 (2.4)	2 (1.2)
**Time between ART start and delivery in days, median (IQR) ^2^**	747 (211–860)	612 (115.5–1822)	175 (110–1594)	557 (110.5–1621.5)
**Children**				
**Nevirapine prophylaxis 0–6 weeks, *n* (%)**				
Yes	7 (77.8)	72 (91.1)	68 (81.0)	147 (85.5)
No	2 (22.2)	7 (8.9)	15 (17.9)	24 (14.0)
Missing data	0	0	1 (1.2)	1 (0.6)

^1^ Abbreviations: TDF: tenofovir, 3TC: lamivudine; EFV: efavirenz, AZT: zidovudine, sdNVP: single dose nevirapine; ^2^ Missing data for 2 in the infected group, 18 in the uninfected and 29 in the indeterminable group.

**Table 5 ijerph-18-02751-t005:** Child cotrimoxazole preventive therapy (CPT) by child HIV status.

Provision of CPT	HIV-Infected	HIV-Uninfected	HIV Status Indeterminable	Total
	*n* = 9	*n* = 79	*n* = 84	*n* = 172
**Started CPT, *n* (%)**	9 (100)	79 (100)	82 (97.6)	170 (98.8)
Age in weeks at start of CPT, median (IQR)	13.1 (10.7–15.1)	6.6 (5.3–8.6)	6.9 (4.9–10)	6.7 (5.0–9.4)
Started later than recommended ^1^, *n* (%)	7 (77.8)	47 (59.5)	53 (64.6)	107 (62.9)
Duration of CPT according to guidelines, *n* (%) ^2^		63 (79.7)	0	63 (79.7)
Record of continued CPT, but inadequate duration, *n* (%)	0	12 (15.2)	3 (3.6)	15 (8.7)

^1^ Started CPT after 6 weeks of age, i.e., after child age of 4–6 weeks as recommended by the Tanzanian guidelines; ^2^ Tanzanian guidelines recommend that CPT should be continued until HIV infection is excluded or until 6 weeks after cessation of breastfeeding. All children living with HIV younger than five years of age should receive CPT regardless of symptoms or CD4 percentage. We only assessed data until age 18 months, at which age none of these children had documented CPT.

**Table 6 ijerph-18-02751-t006:** Infant feeding by child HIV status.

Infant Feeding	HIV-Infected	HIV-Uninfected	HIV Status Indeterminable	Total
	*n* = 9	*n* = 79	*n* = 84	*n* = 172
**Exclusive BF ^1^ from 0–6 months, *n* (%) ^2^**				
Yes	0 (0)	31 (39.2)	16 (19.0)	48 (27.9)
No	3 (33.3)	16 (20.3)	12 (14.2)	31 (18.0)
**Mixed feeding from >6–12 months, *n* (%) ^2^**				
Yes	1 (11.1)	30 (38.0)	5 (6.0)	36 (20.9)
No	0 (0)	29 (36.7)	14 (16.7)	43 (25.0)
**Ceased breastfeeding, *n* (%) ^2^**				
At 13 months	1 (11.1)	20 (25.3)	5 (6.0)	26 (15.1)
Before 13 months	0 (0)	30 (38.0)	10 (11.9)	40 (23.2)
After 13 months	0 (0)	16 (20.3)	8 (9.5)	24 (14.0)

^1^ Abbreviation: BF: breastfeeding; ^2^ Missing data: Exclusive breastfeeding: 6 in the infected group, 32 in the uninfected and 56 in the indeterminable group; Mixed feeding: 8 in the infected group, 20 in the uninfected and 65 in the indeterminable group; Ceased breastfeeding: 8 in the infected, 13 in the uninfected and 60 in the indeterminable group.

**Table 7 ijerph-18-02751-t007:** Associations between antiretroviral therapy (ART), maternal WHO stage, timing of infant HIV PCR test, and maternal CD4 cell count by risk of mother-to-child transmission of HIV (*n* = 88 mother-child pairs).

Maternal and Infant Factors	Total, *n* (%)	Infected, *n* (%)	OR (95% CI) ^1^	*p*-Value
**Maternal ART in pregnancy**				
Yes	78 (88.6)	5 (6.4)	1	
No	10 (11.4)	4 (40.0)	9.7 (2.1, 46.1)	0.004
**Infant ART prophylaxis**				
Yes	79 (89.8)	7 (8.9)	1	
No	9 (10.2)	2 (22.2)	2.9 (0.51, 17.0)	0.228
**Mother’s WHO stage ^2^**				
1–3	52 (73.8)	4 (57.1)	1	
4	16 (26.2)	3 (42.9)	2.8 (0.44, 14.0)	0.217
**Infant first HIV PCR test ^2^**				
Early (<2 months of age)	59 (67.8)	2 (3.4)	1	
Late (≥2 months of age)	28 (32.2)	7 (25.0)	9.5 (1.8, 49.4)	0.007
**Maternal CD4 cell count ^2^**				
<200 cells/mm^3^	6 (10.9)	3 (50.0)	15.3 (2.1, 111)	
≥200 cells/mm^3^	49 (89.1)	3 (6.1)	1	0.007

^1^ Odds ratio (OR) and 95% Confidence Intervals (95% CI) and *p*-value obtained by binary logistic regression; ^2^ Missing data: WHO stage: 2 in the infected group, 18 in the uninfected; First HIV PCR test: 1 in the uninfected group; Maternal CD4 cell count: 3 in the infected group, 30 in the uninfected.

## Data Availability

Restrictions apply to the availability of these data which were obtained for the purpose of the current study and are not publicly available. Data may be made available upon reasonable request with the permission of the National Institute for Medical Research and Haydom Lutheran Hospital, Tanzania.

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
