# Peer review of "Achievements and Challenges in the Prevention of Mother-to-Child Transmission of HIV—A Retrospective Cohort Study from a Rural Hospital in Northern Tanzania"

_ijerph, 2021, doi:10.3390/ijerph18052751_

Round 1
Reviewer 1 Report
The prevention of mother to child transmission of HIV has been highly successful. Therefore, studies of this nature are important to identify gaps and support sustainability of this intervention. The authors found about 10% of infection in children born to HIV infected mothers, show that this was mostly associated with failure of the mothers to take ARV and low CD4+ cell count of the mothers. The report is well written, study limitations are highlighted. Gaps in record keeping critical for intervention to improve the proportion of HIV negative children born from HIV infected mothers were identified.
Minor comments:
- Update the statistics in paragraph one of the introduction (year 2017).
- The data on cotrimoxazole prevention therapy should be discussed in relation to the wellbeing of the study cohort.
Author Response
- Update the statistics in paragraph one of the introduction (year 2017).
RESPONSE: Thank you for your general comments. We referred to the number of infected children in 2017 because the time period overlaps with our study period. We have now provided numbers for 2019. For comparison with global figures of incidence in the age below 15 we have changed Tanzanian data from prevalence to number of new HIV-infections in the age group below 15. However, we have omitted prevalence data for the age group 15-49 as this is less relevant for our study. In responding both to you and the other reviewer the first paragraph of the introduction has been rewritten with updated statistics for year 2019 and reference to the Sustainable Development Goals:
“In 2015, the Sustainable Development Goals set out the ambitious targets of ending the acquired immunodeficiency syndrome (AIDS) epidemic and achieving universal health coverage by 2030 (1). This was followed by the goal of eliminating new human immunodeficiency virus (HIV) infections in children by 2018 (2). Yet, in 2019, 150 000 children under 15 years became infected with HIV. The majority of these infections occurred in sub-Saharan Africa, and in Tanzania, there were an estimated 6 300 cases of new HIV infections in this age group (3)”.
- The data on cotrimoxazole prevention therapy should be discussed in relation to the wellbeing of the study cohort.
RESPONSE: Thank you for raising this issue. We have added a paragraph in the discussion.
“The WHO recommends CPT after week 4–6 to prevent bacterial infections, malaria and Pneumocystis jirovecii pneumonia (24). We found that the majority of HEIs started CPT at a later age, possibly putting the children at risk of severe infections. However, the recommendation to give CPT to HIV-exposed, HIV-uninfected infants is based on very-low-quality evidence (24). A recent study from South Africa suggests that cotrimoxazole prophylaxis provides no benefit after age 6 weeks for HIV-uninfected breastfed infants in countries that are unaffected by malaria (25). Malaria is not highly prevalent in children seen at our hospital, and it is possible that that failure to provide CPT in HIV-uninfected infants may not be of great concern.”
Reviewer 2 Report
I evaluated the manuscript entitled: Achievements and challenges in the prevention of mother to child transmission of HIV - a retrospective cohort study from a rural hospital in northern Tanzania
This is a very interesting manuscript on to assess the occurrence of mother to child transmission of HIV (MTCT) by age 18 months, risk factors for transmission, and the implementation of the national prevention of MTCT (PMTCT) program in a rural hospital in Tanzania. The manuscript has its originality but is full of limitations. In my opinion, the main issue refers to the fact that the authors do not treat as losses those who did not manage to do the correct monitoring / training in the given time. In a cohort, this should be considered a loss as it strongly influences the results.
Furthermore, the fact that it was done in a rural hospital in Tanzania only further limits the results.
In addition, I highlight as minor points:
-The introduction needs comparison. For example: "In Tanzania, the estimated HIV prevalence was 0.4% in children under 15 years of age, and 4.7% in the age group 15–49 years" .... But what is the overall rate? at the global level? at what level are we comparing?
-Contextualize the object of study in the background against the 2030 Development Agenda; sustainable development goals and universal health coverage;
-The objective should be more directive, removing methodological aspects such as "... measures as described in the Tanzanian national guidelines".
-The authors abuse acronyms not known internationally. This is not good for a Mega Journal like this.
-Results and discussion should focus on the goal. Points outside of that, just make the text repetitive;
-The limitations can be more honest and point out the factors already pointed out.
Author Response
- This is a very interesting manuscript on to assess the occurrence of mother to child transmission of HIV (MTCT) by age 18 months, risk factors for transmission, and the implementation of the national prevention of MTCT (PMTCT) program in a rural hospital in Tanzania. The manuscript has its originality but is full of limitations. In my opinion, the main issue refers to the fact that the authors do not treat as losses those who did not manage to do the correct monitoring / training in the given time. In a cohort, this should be considered a loss as it strongly influences the results.
RESPONSE: Thank you for pointing out the importance of accounting for “losses”. We agree fully, - but in our opinion we have taken care to point this out both in tables and in text. We have divided HIV-exposed infants into three groups. HIV-infected, HIV-uninfected and HIV status indeterminable. Our main analysis is based on study subjects on whom we have adequate monitoring to determine outcome, i.e. the first two groups, and we have attempted to point this out clearly in the text (section 2.3): “The children were followed until age 18 months and were finally categorized into one of three groups: “HIV-infected”, “HIV-uninfected”, and “HIV status indeterminable”. Reasons for being included in the latter group included no or inadequate testing (in relation to birth and breastfeeding), transfer out, loss to follow-up, and death without appropriate HIV testing».
The last group (HIV status indeterminable), which we presume you refer to as “losses”, comprises participants on whom we have inadequate data to determine outcome. In order to more clearly point out the definition of transfer out and loss to follow-up we have added a few words (underlined below) in the manuscript.
“Reasons for being included in the latter group included no or inadequate testing (in relation to birth and breastfeeding), transfer out (to other health care providers), loss to follow-up (for unknown reason), and death without appropriate HIV testing”.
Even though we have incomplete data in this group it is an important result that almost one-half of the children could not be adequately accounted for with respect to HIV-status at age 18 months. This is related to our third aim: i.e. the implementation of PMTCT measures as described in the Tanzanian national guidelines. In our opinion there is merit to reporting in detail what characterized this group as compared to the two others as this may serve to identify areas for improvement in the PMTCT program.
- Furthermore, the fact that it was done in a rural hospital in Tanzania only further limits the results.
RESPONSE: We agree that the rural environment may explain some of the limitations of this study. For example, poverty and challenges related to transportation may at least in part explain why patients were transferred out or were lost to follow-up. However, according to the world bank, 65.5% of the population of Tanzania was living in rural areas in 2019. Therefore, it is our opinion that the rural environment of this study is also a strength, adds value compared to studies that are conducted in urban areas, and demonstrates challenges that are very relevant to a large section of the population of Tanzania.
We have already tried to highlight this in the last paragraph of the discussion.
We have replaced the following paragraph in the discussion
“Of all children enrolled in our study, one third (32.0%) were lost to follow up and one-eighth (12.7%) were transferred out, largely explaining why we were unable to adequately determine the HIV status at age 18 months in almost one half of exposed children. Loss to follow-up has also been identified as a significant challenge in several other studies from Africa [15,19,21,26,27].”
with further explanations on how the groups “lost to follow-up” and “transfer out” may relate to the rural environment.
“Of all children enrolled in our study, one third (32.0%) was lost to follow-up for unknown reasons, and one-eighth (12.7%) was transferred out to other health care providers. This largely explains why we were unable to adequately determine the HIV status at age 18 months in almost one half of the exposed children. Long distances, cost of travel and poor communications in our rural area may be good reasons for transfer of patients to health care providers nearer to the home. Loss to follow-up for unknown reasons is a greater cause for concern and has also been identified as a significant challenge in several other studies from Africa (13, 17, 19, 28, 31). In addition to the above factors, stigma associated with HIV is a likely explanation for the high rate of loss to follow-up”.
In addition, I highlight as minor points:
RESPONSE: We appreciate these minor points and have attempted to respond to these.
- -The introduction needs comparison. For example: "In Tanzania, the estimated HIV prevalence was 0.4% in children under 15 years of age, and 4.7% in the age group 15–49 years" .... But what is the overall rate? at the global level? at what level are we comparing?
RESPONSE: Thank you for this and the following comment relating to the introduction. We respond to this and the following comment under comment 4.
- -Contextualize the object of study in the background against the 2030 Development Agenda; sustainable development goals and universal health coverage;
RESPONSE: In responding both to you and the other reviewer, the first paragraph of the introduction has been rewritten with updated statistics for year 2019 and reference to the Sustainable Development Goals. For comparison with global figures of incidence in the age below 15 we have changed Tanzanian data from prevalence to number of new HIV-infections in the age group below 15. However, we have omitted prevalence data for the age group 15-49 as this is less relevant for our study.
This is the new first paragraph:
“In 2015, the Sustainable Development Goals set out the ambitious targets of ending the acquired immunodeficiency syndrome (AIDS) epidemic and achieving universal health coverage by 2030 (1). This was followed by the goal of eliminating new human immunodeficiency virus (HIV) infections in children by 2018 (2). Yet, in 2019, 150 000 children under 15 years became infected with HIV. The majority of these infections occurred in sub-Saharan Africa, and in Tanzania, there were an estimated 6 300 cases of new HIV infections in this age group (3)”.
- -The objective should be more directive, removing methodological aspects such as "... measures as described in the Tanzanian national guidelines".
RESPONSE:
We are unsure what you mean by “more directive” and we find it inappropriate to change or remove any of our three pre-defined aims that can be found in the last paragraph of the introduction (numbering only added below) i.e. to assess
- the occurrence of MTCT by age 18 months
- the influence of known predictors on transmission
- the implementation of PMTCT measures as described in the Tanzanian national guidelines
The last of these aims is closely linked to the first two and was a major motivating factor for this study. This is explained in the previous paragraph:
“Although PMTCT activities are in place in most resource-constrained countries, there is a need for studies that examine the level of implementation and quality of the documentation at the health facility level, and how this may affect the reported rate of MTCT”.
- -The authors abuse acronyms not known internationally. This is not good for a Mega Journal like this.
RESPONSE: We agree that we use many acronyms and abbreviations but many of these are hard to avoid without tedious and long sentences and words. Most of these are very well known to general readers, e.g. AIDS, HIV, DNA, WHO and we presume that there is no objection to their use. Some are less well known outside the field of HIV but have been kept after initial explanations as they are in frequent use by WHO and other international agencies.
We have removed the following abbreviations: HLH (Haydom Lutheran Hospital, KCMC (the Kilimanjaro Christian Medical Centre), NVP (nevirapine prophylaxis) (NVP used in the tables with explanation in the legend), HEID (HIV early infant diagnosis) except in Figure 2 where HEID is used with explanation in the legend.
- -Results and discussion should focus on the goal. Points outside of that, just make the text repetitive;
RESPONSE: We regret that you find the text repetitive. In our opinion, our results and discussion are related to the aims. For instance, our text mainly refers relatively briefly to results that are shown in tables. Tables 1 refers to laboratory testing of infants (related to aim 1 and 3), table 2 and 3 shows characteristics of mothers and infants respectively (related to aim 1,2 and 3), table 4 and 5 show pharmacological interventions (relevant to aims 1,2 and 3) and table 6 shows breast-feeding data (relevant to aims 1, 2 and 3). Finally, table 7 shows risk factors associated with PMTCT (aim 2). In the discussion, we have only reiterated main findings that are of high relevance to our aims and have discussed own findings in light of other studies. We have reorganized the discussion to some extent and hope you find this lay-out less repetitive, more informative, and more clearly linked to our aims.
If there are some parts that you find clearly superfluous, please specify in order that we can address them.
- -The limitations can be more honest and point out the factors already pointed out.
RESPONSE: Please see our answer to your previous comments 1 and 2
Additional comments from us:
We have added a sentence (number two) in the conclusion to better align this with all aims.
The conclusion has been changed from:
“During 2014–2016, one in ten HIV-exposed children with adequate follow-up were infected from their mothers by age 18 months. Almost one-half of the children enrolled at RCHS could not be adequately assessed for HIV infection, largely due to loss to follow-up and transfer out.
Our study demonstrates a need for enhanced efforts to prevent and diagnose MTCT, reduce loss to follow-up of HIV-exposed children, to improve documentation in registries and records, and to facilitate ease of linkage between these”.
To:
“One in ten HIV-exposed children with sufficient data for assessment were infected from their mothers by age 18 months. Risk of infection was associated with low maternal CD 4 cell counts, mothers not taking ART during pregnancy, and late presentation of infants for HIV testing. Almost one-half of the children enrolled at RCHS could not be adequately assessed for HIV infection, largely due to loss to follow-up and transfer out to other health care providers.
Our study demonstrates a need for enhanced efforts to prevent and diagnose MTCT, reduce loss to follow-up of HIV-exposed children, to improve documentation in registries and records, and to facilitate ease of linkage between these”.
In addition to the above-mentioned changes, we have also made some minor changes in the manuscript, including moving some paragraphs in the text to improve language and improve the discussion.
Some superfluous references have been removed.
Round 2
Reviewer 2 Report
All suggestions were duly incorporated